# On the Question of the Complex Processing of Pyrite Cinders

**Bagdaulet Kenzhaliyev, Tatiana Surkova, Dinara Yessimova \*** [ID]**, Yerkezhan Abikak** [ID]**, Ainur Mukhanova
and Dametken Fischer** [ID]

JSC "Institute of Metallurgy and Ore Beneficiation", Satbayev University, Shevchenko str., 29/133,
Almaty 050010, Kazakhstan

\* Correspondence: dina-28@inbox.ru

**Abstract:** A complex processing variant for pyrite cinders, i.e., the technogenic waste generated in the production of sulfuric acid, was proposed. This method provided preliminary chemical activation of the initial raw materials that comprised thermal treatment with a sodium bicarbonate solution and resulted in structural and phase changes of separate minerals. Due to chemical activation, it was possible to separate the nonferrous metals into separate products (in addition to the partial extraction of iron) and then concentrate the noble metals in the residue. The noble metals were then able to be extracted through a leaching process with a complex reagent based on sulfur compounds and subsequent cementation with zinc dust. The developed method, unlike pyrometallurgical methods, is less energy-consuming and more easily implemented than the known hydrometallurgical variants, enabling the separation of nonferrous metals and the partial separation of iron into separate middlings at the first stage. Noble metals are concentrated in the residue and extracted from it.

**Keywords:** hydrothermal treatment; phase composition; valuable components; complex reagent based on sulfur compounds; oxidizers





## 1. Introduction

At present, according to data obtained by various experts, 14 to 25 billion tons of technogenic waste is accumulated in the Republic of Kazakhstan, and less than 2% of this is processed [1]. The accumulation of technogenic waste has a detrimental effect on the ecology of the surrounding land and also results in the rejection of significant land areas, causing adverse hydrogeological and geochemical changes. Therefore, the issue of waste utilization has become increasingly acute.

Anthropogenic waste also includes pyrite cinders from pyrite concentrate roasting in the production of sulfuric acid. Pyrite cinders comprise finely dispersed solid waste with a complex structure based on iron compounds (approximately 60%). The composition of pyrite cinders also includes nonferrous and noble metals [2].

It is known from the literature that several of the methods used to process these raw materials are based on the pyrometallurgical processes of chlorination and chloride sublimation [3,4]. The chlorination method involves low-temperature (550–600 °C) roasting with common salt or calcium chloride, and was first introduced at the Duisburg Plant in Germany [4]. The most promising direction of chlorination roasting is chloride sublimation [3]. In this method, granular raw material is loaded into a shaft furnace heated by hot (1250 °C) gases. In an oxidizing atmosphere, iron is not chlorinated, and nonferrous metal impurities are removed in the form of chlorides, which have low boiling points. Through this, there is a separation of nonferrous metals from iron [4,5]. This process was carried out in Finland at a plant in Imatra. This method favorably differs from the low-temperature chlorination roasting of cinder and is characterized by sufficiently high rates of extraction of nonferrous metals. The disadvantages of the above methods include increased energy consumption and the lack of complexity in the extraction of valuable components.

Of the chloride-free methods, the variant developed by Outokumpu involving the smelting of pyrite concentrates in a neutral atmosphere with sulfur sublimation deserves the most attention. This method makes it possible to obtain a product containing up to 67% iron but does not provide for the extraction of nonferrous and precious metals [6].

Of the hydrometallurgical options, one particular method has proven itself well, which includes a sequential four-stage leaching of the feedstock. Initially, nonferrous metals are leached with a solution of sulfuric acid, and then noble metals are leached with a hydrochloric acid solution of thiocarbamide [7]. The disadvantages of this method include its multiple stages and the low levels of iron and precious metals extracted.

Among the more promising methods, one should also note the method of deep processing of pyrite cinders [8], including the leaching of nonferrous metals by a bacterial complex. However, the long duration of bacterial leaching and the complexity of the process of cultivating acidophilic thionic bacteria reduce its value.

It should be noted that a significant amount of pyrite cinders has been accumulated not only in Kazakhstan but also in other countries. In particular, in Russia, the amount of unutilized pyrite cinders exceeds 30 million tons. Hydrometallurgical methods remain the predominant direction of research in the processing of pyrite cinders in Russia [9–11]. One of the latest options proposed by researchers is flotation enrichment. This method allows the extraction of 60% copper and 30–40% gold. The output of the concentrate is 5–8% [12].

A detailed study on processing pyrite cinders allowed Chinese scientists to develop a method that involved reductive roasting–leaching–magnetic separation. Maximum extraction of copper during leaching was achieved (82.18%), and a high-quality iron concentrate with a content of 65.58% Fe and 0.17% Cu was obtained from the residue via magnetic separation [13].

Thus, it follows from the foregoing that there are a significant number of options for processing pyrite cinders, but not all of them provide for the extraction of precious metals, which are the most valuable components of raw materials.

In the hydrometallurgical extraction of precious metals, one of the following methods is usually used: cyanidation, hydrochlorination, or thiosulfate opening [14–20]. Cyanidation, as a rule, is preceded by a number of technological methods, which are reduced mainly to the preliminary oxidation of raw materials. The advantage of hydrochlorination is the combination of two processes at once: oxidation and leaching. Along with cyanidation and hydrochlorination, increasing attention in the global practice of precious metal extraction is given to thiosulfate processing [14,15].

The noble metal leaching method, using ammonium or sodium thiosulfate solutions in the presence of sulfite ions and copper ions in an alkaline medium, is a well-known global practice. It is used to process cuprous, manganese, carbonaceous, and other ores unsuitable for the leaching process with cyanide solutions. Over time, the list of raw materials has expanded significantly due to the systematic research of Canadian, Chinese, and American scientists [16].

A large amount of work on thiosulfate leaching of precious metals has been carried out by Russian researchers at the Irgiredmet Institute. This process has been tested using many types of raw mineral materials, including quartz gold–silver and clay-based gold-bearing ores, as well as flotation tailings from a number of deposits [17].

In Kazakhstan, the first studies on thiosulfate technologies for extracting gold from ores were carried out in the 1990s. The object of this research was carbonaceous arsenic sulfide ore. Later, studies were carried out on the industrial development of thiosulfate technology in relation to gold–copper sulfide ores and gold-bearing weathering crusts. [18–20].

Work aimed at improving thiosulfate technology continues today. In order to ensure overall technological safety, it is necessary to propose new, more efficient combined processing methods for each individual type of raw material, particularly for those using oxidizers, taking into account the characteristics and composition of the raw material considered.

## 2. Results and Discussion

The JSC "Institute of Metallurgy and Enrichment" developed a method for the complex processing of pyrite cinders based on the chemical activation of technogenic raw materials [2]. The method comprises the thermal processing of raw materials with a solution of sodium bicarbonate and achieves, at the first stage, the separation of nonferrous metals and partial separation of iron into separate middlings and concentrates noble ones from the remaining material. Chemical activation is preceded by magnetic separation, which allows useful components to be concentrated in a magnetic fraction. This magnetic fraction is subjected to chemical activation.

The optimal conditions for chemical activation were as follows: temperature, 120 °C; duration, 90 min; L:S ratio, 4:1; concentration of sodium bicarbonate solution, 60 g/dm$^3$. Under these conditions, the maximum changes in the phase and chemical composition of the raw material were obtained. X-ray patterns of the feedstock before and after chemical activation, as well as the phase composition, are presented in Figures 1 and 2, and in Tables 1 and 2.

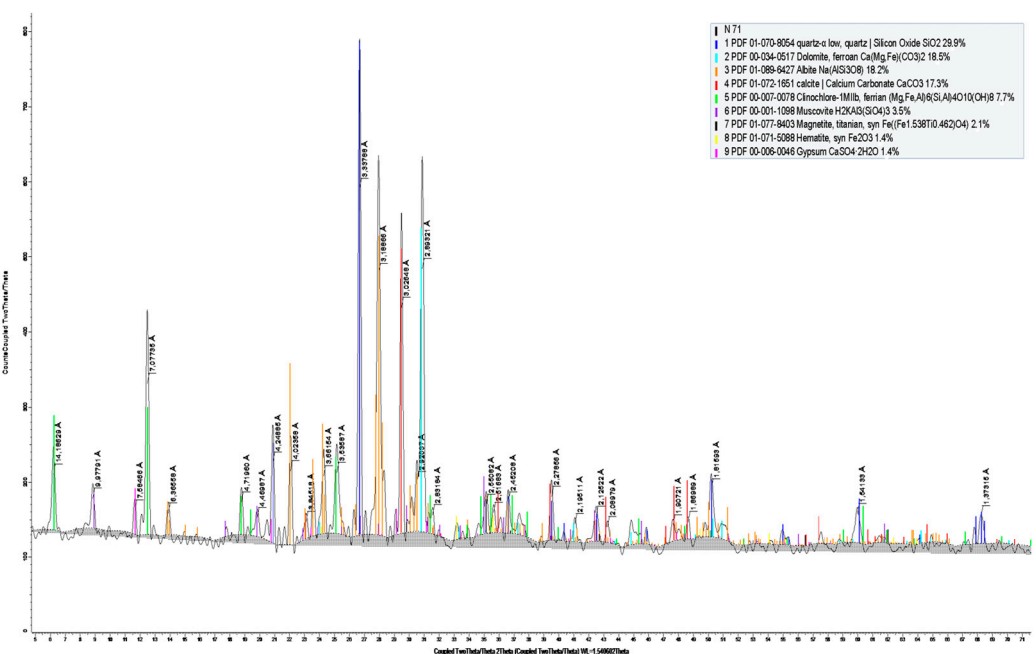

**Figure 1.** X-ray pattern of the initial sample of pyrite cinders.

**Table 1.** Phase composition of the initial sample of pyrite cinders.

| Name | Formula | % |
|---|---|---|
| Magemite | $Fe_2O_3$ | 25.1 |
| Hematite | $Fe_2O_3$ | 19.1 |
| Quartz | $SiO_2$ | 18.0 |
| Albite | $Na(AlSi_3O_8)$ | 10.2 |
| Trisodium phosphate zinc oxide hydrate | $Na_3Zn_4O(PO_4)3(H_2O)_6$ | 9.5 |
| Sodium aluminum silicate | $NaAl_3Si_3O_{11}$ | 6.7 |
| Barium ferrite | $BaFe_2O_4$ | 4.7 |
| Natrozharozit | $(Na_{0.67}(H_3O)_{0.33})Fe_3(SO_4)_2(OH)_6$ | 4.2 |
| Dolomite | $CaMg(CO_3)_2$ | 2.5 |

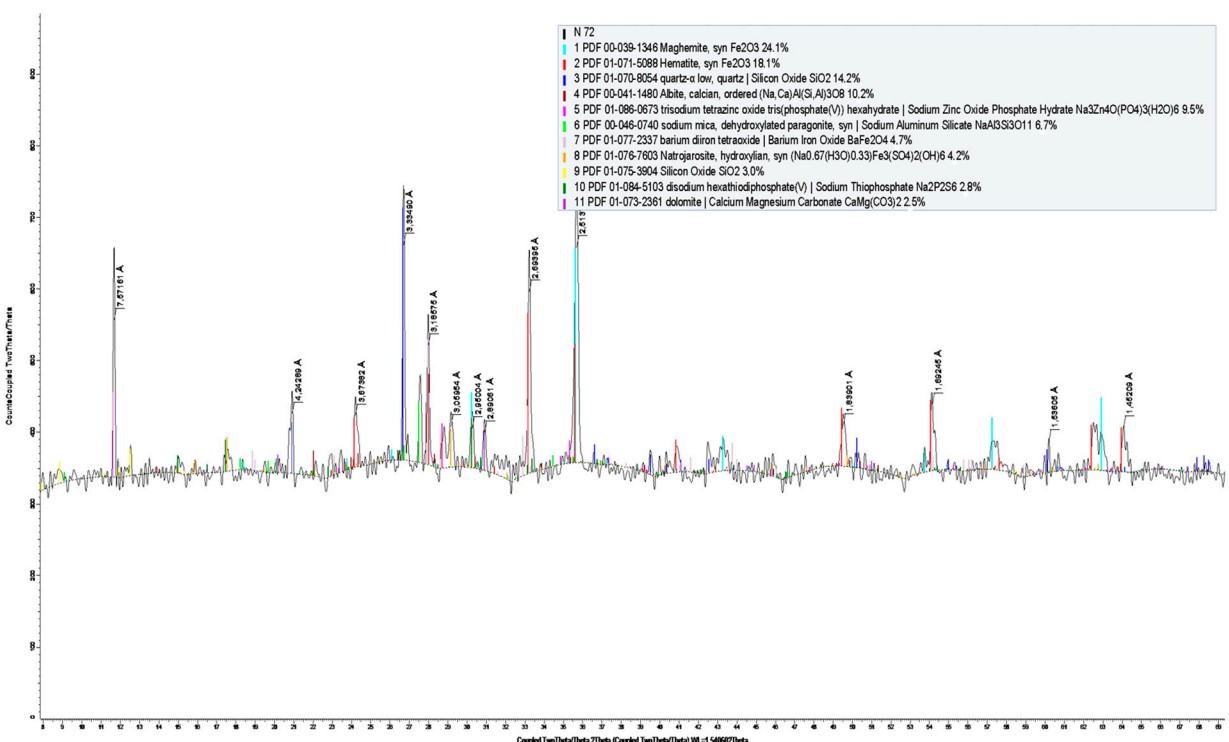

**Figure 2.** X-ray pattern of pyrite cinders after thermochemical activation.

**Table 2.** Phase composition of pyrite cinders after thermochemical activation.

| Name | Formula | % |
|---|---|---|
| Magemite | $Fe_2O_3$ | 30.7 |
| Hematite | $Fe_2O_3$ | 21.7 |
| Quartz | $SiO_2$ | 13.3 |
| Albite | $Na(AlSi_3O_8)$ | 5.8 |
| Sodium aluminum silicate | $NaAl_3Si_3O_{11}$ | 7.8 |
| Barium ferrite | $BaFe_2O_4$ | 7.0 |
| Natrozharozit | $(Na_{0.67}(H_3O)_{0.33})Fe_3(SO_4)_2(OH)_6$ | 4.8 |
| Sodium thiophosphate | $Na_2P_2S_6$ | 3.5 |
| Magnesium aluminum silicate | $(MgAl_2Si_3O_{10})_{0.6}$ | 2.4 |
| Calcium silicate | $CaSiO_3$ | 1.8 |

As a result of the hydrothermal treatment, the dolomite phase disappeared. Magnesium passed into a new aluminosilicate phase $(MgAl_2Si_3O_{10})_{0.6}$. The albite phase was almost halved; as a result of thermochemical activation, when interacting with a solution of sodium bicarbonate, the phases of the waste rock were dissolved. The waste rock phases covered and cemented the nonferrous minerals. As a result of thermochemical activation, when interacting with a solution of sodium bicarbonate, they dissolved. This contributed to more efficient leaching of the raw material components by sulfuric acid. The acid concentration was 15–20% when the process was carried out at a temperature of $-60\ °C$. Under these conditions, wt. %: CuO 76.8, ZnO 75.9, and $Fe_2O_3$ 26.0 was extracted. When leaching raw materials without chemical activation, the degree of extraction of nonferrous metals was 15–20% lower.

Such a transformation in the process of chemical activation facilitates the further transformation of compounds of ferrous, nonferrous, and noble metals into final prod-

ucts, contributing to the development of a more compact technology for processing this raw material.

The technological scheme for the processing of pyrite cinders is shown in Figure 3.

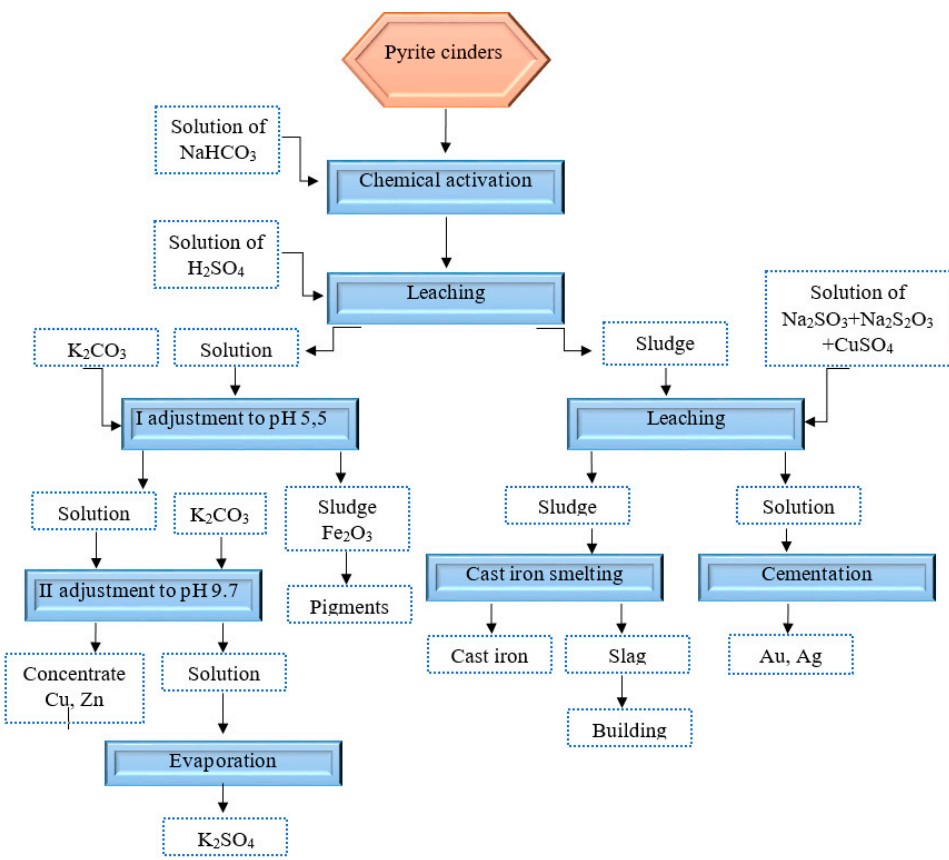

**Figure 3.** Technological scheme of pyrite cinders processing.

As stated above, the most valuable components of pyrite cinders were noble metals concentrated in the residue after the extraction of nonferrous metals and partial extraction of iron. A polished section was formed for mineralogical analysis of this material. Its study under a microscope showed that hematite and goethite–hydrogoethite were dominant among the ore minerals in the residue. Hematite was represented as fine-grained zonal aggregates, indicative of the oxidation stage. Much of the magnetite was subject to martitization, wherein hematite in the form of thin plates pseudomorphically replaces magnetite along the periphery, cracks, and voids (Figure 4).

Table 3 presents the mass fraction data for each mineral.

**Table 3.** Mass fraction of minerals in the residue.

| Minerals | The Ratio of Minerals in the Residue, % |
|---|---|
| Pyrite | 9.2 |
| Gold | 0.3 |
| Chalcopyrite | 0.1 |
| Silver | 9.4 |
| Hematite | 30.6 |
| Cerussite | 1.3 |
| Magnetite | 5.8 |
| Goethite–hydrogoethite | 43.3 |

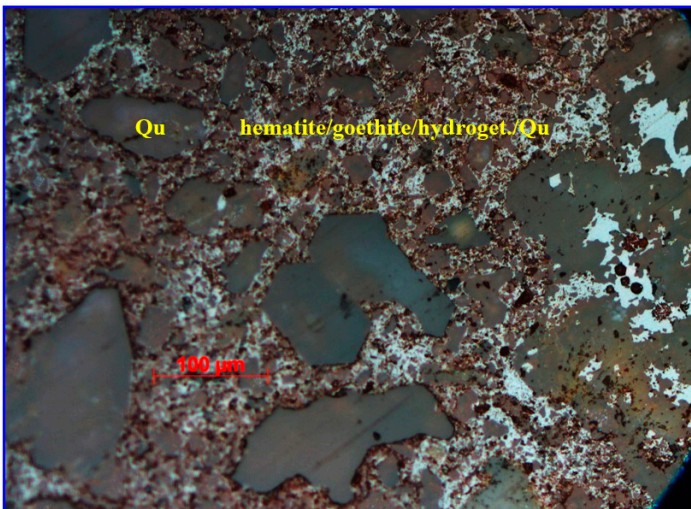

**Figure 4.** A section of the briquette residue after thermochemical activation and extraction of nonferrous metals with an area of S = 500 μm × 700 μm. Magnification × 200 times.

The table shows that sulfide mineralization was extremely sparse and was represented by pyrite and chalcopyrite.

The gold content in the residue was 1.9 g/t, and the silver was 16.9 g/t.

The gold in the residue was in ultra-fine- and fine-dispersed states, both were free and in aggregates with hematite and goethite–hydrogoethite (Table 4). The size of the gold particles varied from 0.5 to 1.4 μm. The shapes of gold grains were different: spherical, irregular, hooked, and less often elongated. The surface of the grains was both smooth with clear contours as well as rough. The color varied from pale yellow to bright gold. Additionally, there were gold particles in a free state, covered with oxidation films of a goethite–limonite composition called "rusty gold" [21].

**Table 4.** Forms of gold in the remaining material.

| Forms of Gold | % |
| --- | --- |
| Gold in the free state | 58.33 |
| Gold in association with hematite | 25.0 |
| Gold in association with goethite–hydrogoethite | 16.67 |
| «Rusty gold» | 33.33 |

Silver occurs in the form of free ultra-fine and pulverized particles of round, elongated, and irregular shapes ranging in size from 0.4 μm to 5.5 μm, as well as in intergrowths with goethite–hydrogoethite. In isolated cases, they can reach up to 9.3 × 16.1 μm in size.

Thus, from the presented results, it follows that the basis of the residue in which noble metals are concentrated is in the form of iron compounds.

Factors, such as the temperature, concentration of thiosulfate, ammonia, divalent copper ions, and sulfite ion, as well as the ore grinding degree, influence the leaching process [22–29].

Researchers established that the silver dissolution process is more sensitive to temperature changes as compared to that of gold. At the same time, the gold dissolution rate was significantly inhibited by the products obtained due to the interaction of divalent copper ions with thiosulfate [22,23].

The optimum thiosulfate concentration in the leaching solution varied widely and depended on the ore material composition. An increase in the extraction degree of noble metals was noted with an increase in thiosulfate concentration in the solution but, in turn,

led to an increase in its consumption. The optimum concentration was determined as a compromise between the leaching rate and thiosulfate consumption [22,23].

Quite a different dependence of noble metal extraction rate was observed when the ammonia concentration was increased. The leaching rate decreased in this case. Two factors influenced the process: film passivation by oxygen–hydrate coating of gold and a reduction in the copper recovery rate. Therefore, the ammonia concentration was chosen as a compromise between the gold leaching rate and copper recovery rate [24,25].

As for the concentration of copper ions, the noble metal leaching rate does not depend on the initial concentration of copper. At the same time, as noted above, the reaction products that were formed during the reduction of divalent copper ions by thiosulfate interfere with gold oxidation and inhibit the leaching process [26,27].

The function of the sulfite ion introduced into the leaching solution is to stabilize the thiosulfate. In the absence of sulfite, thiosulfate disintegrates to form elemental sulfur, which covers the minerals, preventing the extraction of noble metals [28].

In addition, as specified by a number of researchers, the noble metal leaching process is influenced by the ratio of areas of the mineral and the associated gold, i.e., the ore grinding degree. The optimal grain size of the mineral should be five times the size of the associated gold [29].

Thus, choosing optimum concentrations of the components included in the complex reagent, along with the process temperature and degree of ore grinding, plays a decisive role in the leaching process for noble metals.

Based on an analysis of the scientific literature [22–31], the following compositions of the complex reagent were selected to process the residue obtained as a result of the preliminary activation of pyrite cinders and the extraction of nonferrous metals (Figure 1). The conditions of the noble metal leaching process were studied (Tables 5 and 6). In the course of the research, it was found that the optimal degree of grinding of raw materials is 0.074 mm.

**Table 5.** Leaching process results for residue after thermochemical activation based on sulfur compounds.

| № | Component Concentrations, g/dm$^3$ | S:L | Temperature, °C | Time, h | Extraction, % | |
|---|---|---|---|---|---|---|
| | | | | | Au | Ag |
| 1 | c (Na$_2$SO$_3$)—40; c (NH$_4$OH)—15; c (CuSO$_4$)—8. | 1:5 | 20 | 4 | 20 | 7.9 |
| 2 | c (Na$_2$SO$_3$) = 80; c (NH$_4$OH) = 15; c (CuSO$_4$) = 8 | 1:5 | 20 | 4 | 18 | 7.9 |
| 3 | c (Na$_2$SO$_3$) = 40; c (NH$_4$OH) = 15; c (CuSO$_4$) = 8 | 1:5 | 60 | 4 | 35 | - |
| 4 | c (Na$_2$SO$_3$) = 100; c (Na$_2$S$_2$O$_3$) = 50; c (CuSO$_4$) = 2.5 | 1:10 | 20 | 4 | 65.12 | 11.84 |
| 5 | c (Na$_2$SO$_3$) = 100; c (Na$_2$S$_2$O$_3$) = 50; c (CuSO$_4$) = 2.5 | 1:10 | 20 | 4 | 56.51 | 10.50 |

**Table 6.** Leaching process results for residue after thermochemical activation with complex reagents based on sulfur compounds.

| № | Component Concentrations, g/dm$^3$ | S:L | Temperature, °C | Time, h | Extraction, % | |
|---|---|---|---|---|---|---|
| | | | | | Au | Ag |
| 1 | c (Na$_2$SO$_3$) = 100; c (Na$_2$S$_2$O$_3$) = 50; c (CuSO$_4$) = 2.5 | 1:10 | 20 | 7 | 72 | 55 |
| 2 | c (Na$_2$SO$_3$) = 40; c (NH$_4$OH) = 15; c (CuSO$_4$) = 8 | 1:5 | 20 | 7 | 70 | 45 |
| 3 | c (Na$_2$S$_2$O$_3$) = 80; c (NH$_4$OH) = 15; c (CuSO$_4$) = 8 | 1:5 | 20 | 7 | 62 | 48 |
| 4 | c (Na$_2$S$_2$O$_3$) = 80; c (NH$_4$OH) = 15; c CuSO$_4$) = 8 | 1:5 | 60 | 7 | 67 | 42 |
| 5 | c (Na$_2$SO$_3$) = 100; c (Na$_2$S$_2$O$_3$) = 50; c (CuSO$_4$) = 2.5 | 1:10 | 60 | 7 | 68.4 | 90.8 |
| 6 | c(Na$_2$SO$_3$) = 100; c (Na$_2$S$_2$O$_3$) = 50; c (CuSO$_4$) = 2.5 | 1:10 | 20 | 7 | 67.1 | 98.5 |
| 7 | c (Na$_2$SO$_3$) = 100; c (Na$_2$S$_2$O$_3$) = 50; c (CuSO$_4$) = 2.5 | 1:10 | 60 | 7 | 87.2 | 75.1 |
| 8 | c (Na$_2$SO$_3$) = 100; c (Na$_2$S$_2$O$_3$) = 50; c (CuSO$_4$) = 2.5 | 1:10 | 20 | 7 | 67.3 | 87.2 |

The results of the studies show that the maximum gold extraction degree was 87.2%, and the maximum silver extraction degree was 75.1% when the following complex reagent composition was used, and is listed in $g/dm^3$: $Na_2SO_3$, 100; $Na_2S_2O_3$, 50; $CuSO_4$, 2.5 at a temperature of 60 °C and process time of 7 h (Table 6, experiment No. 7).

For further research, the composition of the solution of experiment No. 7 (Table 6) was chosen, and the effect of the oxidizing agents [32] on the leaching of precious metals was studied. The research results are presented in Tables 7 and 8.

**Table 7.** Results of leaching of residue after thermochemical activation in the presence of sodium hypochlorite.

| Leaching Conditions | S:L | Temperature, °C | Time, h | Extraction, % | |
|---|---|---|---|---|---|
| | | | | Au | Ag |
| Stage I leaching Leaching with 15% sodium hypochlorite solution, PH = 9.5 | 3:1 | 20 °C | 4 | - | - |
| Stage II leaching c ($Na_2SO_3$) = 100 $g/dm^3$; c ($Na_2S_2O_3$) = 50 $g/dm^3$; c($CuSO_4$) = 2.5 $g/dm^3$; PH = 9.5 | 10:1 | 20 °C | 7 | 84.1 | 36.1 |

**Table 8.** Results of leaching of residue after thermochemical activation in the presence of manganese dioxide.

| Name | pH | S:L | Time, h | Concentration $MnO_2$, in Solution, $g/dm^3$. | E, % Au | E, % Ag |
|---|---|---|---|---|---|---|
| Leaching of pyrite cinders with $MnO_2$ | 9.5 | 1:10 | 7 | 5 | 65.53 | 31.04 |
| Leaching of pyrite cinders with $MnO_2$ | 9.7 | 1:10 | 7 | 10 | 90.1 | 59.5 |
| Leaching of pyrite cinders with $MnO_2$ | 9.4 | 1:10 | 7 | 15 | 71.2 | 27.14 |

The results show that no leaching of precious metals was observed at the first stage. In the second stage, 84.1% of gold and 36.1% of silver were recovered.

As follows from the table, the optimum flow rate of the oxidizer should be considered as 10 $g/dm^3$. A further increase in the flow rate was not optimal because it reduced the effectiveness of the leaching process.

Thus, the conducted research showed the expediency of using complex reagents in the following composition for the noble metal leaching process listed in $g/dm^3$: $Na_2SO_3$, 100; $Na_2S_2O_3$, 50; $CuSO_4$, 2.5. The gold extraction degree was 87.2, and that of silver was 75.1% over 7 h. The use of an oxidizer—manganese dioxide—allowed the amount of gold extracted to reach 90.1%, even at room temperature.

Cementation with zinc dust is an effective way of extracting gold and silver from thiosulfate solutions. In this process, gold and silver are reduced, and zinc is dissolved to form a complex with a thiosulfate or ammonium ion.

Gold extraction with zinc dust can be represented by the following reactions:

$$Zn + 2Au(S_2O_3)_2{}^{3-} \longrightarrow 2Au + Zn(S_2O_3)_2{}^{2-} + 2\,S_2O_3{}^{2-}$$

$$Zn + 2Au(S_2O_3)_2{}^{3-} + 4NH_3 \longrightarrow 2Au + Zn(NH_3)_4{}^{2+} + 4\,S_2O_3{}^{2-}$$

There is a reduction of copper ions, first to the monovalent state and then to metallic copper, as well as sulfite sulfate thiosulfate, along with the reduction of noble metals. Consumption of the cementitious reagent—zinc for the reduction of copper and sulfur—significantly exceeds the consumption of the gold and silver reduction.

The precipitation of precious metals was carried out with zinc dust from the productive solution. The content of the precious metals in the solution was as follows, in $g/dm^3$: gold,

0.177; silver, 0.12. The precipitate was analyzed using an electron microscope. The results obtained are presented in Figure 5 and Table 9.

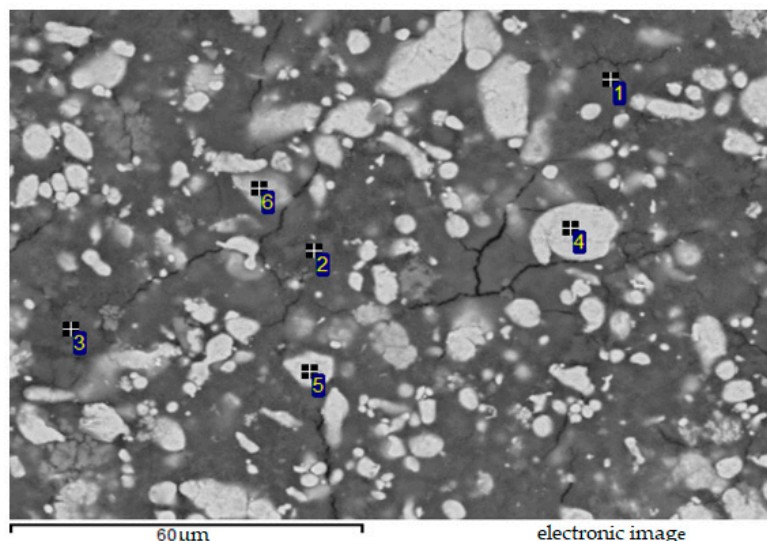

**Figure 5.** The results of electron microscopic study of the sediment. Magnification by 60 μm.

**Table 9.** Elemental composition of the sediment.

| Spectrum | O | Si | S | Fe | Cu | Zn | Ag | Au |
|---|---|---|---|---|---|---|---|---|
| 1 | 37.98 | 11.11 | 18.36 | 0.05 | 2.52 | 29.24 | 0.00 | 0.74 |
| 2 | 46.18 | 17.33 | 19.92 | 0.32 | 1.01 | 14.63 | 0.02 | 0.58 |
| 3 | 37.01 | 7.74 | 24.40 | 0.03 | 17.52 | 12.36 | 0.01 | 0.93 |
| 4 | 8.30 | 2.33 | 0.56 | 0.03 | 0.42 | 88.30 | 0.02 | 0.05 |
| 5 | 3.05 | 0.45 | 0.14 | 0.07 | 0.97 | 95.28 | 0.03 | 0.00 |
| 6 | 18.51 | 8.09 | 1.36 | 0.02 | 0.51 | 71.45 | 0.04 | 0.02 |

The table shows the composition of the individual zones of the sediment. The prevailing elements were sulfur, silicon, copper, and iron. The significant oxygen content indicates the presence of elements, mainly in the form of oxides.

Thus, the fundamental possibility of extracting precious metals by leaching with a complex reagent based on sulfur compounds and a subsequent reduction with zinc dust has been shown.

## 3. Materials and Methods

### 3.1. Research Methods

A sieve analyzer (VP-30, Vibrotechnik, St. Petersburg, Russia) with a vibration frequency of 1500 rpm and vibration amplitude of 1 mm was used for magnetic separation, and a thermostatically controlled unit with 4 autoclaves rotating through the head with a working volume of 250 cm$^3$ each were used for the chemical activation of magnetic fractions. The magnetic separation of the sample was carried out at a magnetic field strength of 200–400 oersteds.

Magnetic separation of the sample was carried out at a magnetic field strength of 200–400 oersteds, chemical activation was carried out at temperatures in the range of 90–230 °C, and the sodium bicarbonate concentration was in the range of 40–120 g/dm$^3$. The maximum content in the sodium bicarbonate solution of 120 g/dm$^3$ was chosen, taking into account its solubility limit. The T:L ratio varied between 2.0–10.0, and the

duration of the process varied from 30 to 300 min. The optimal conditions for the process were determined.

Leaching of the activated cake in order to extract nonferrous metals and, to a lesser extent, iron was carried out in a thermostated reactor with sulfuric acid at a concentration of 15%, a ratio of T:L equal to 1:3, and a temperature of 60 °C. In the process of leaching, the pulp was mixed with a PE 8399 mixer from Ekros. The leaching of noble metals was carried out at room temperature with a complex reagent based on sulfur compounds at different ratios of the initial components.

The T:L ratios were 1:5 and 1:10, and the leaching times were 4 and 7 h.

When studying the effect of the oxidizing agents on the degree of leaching of noble metals, sodium hypochlorite and manganese dioxide were used as oxidizing agents. When using sodium hypochlorite, leaching was carried out in two stages. In the first stage, the residue after the extraction of nonferrous metals and the partial extraction of iron was leached with a solution of sulfuric acid at a concentration of 10 $g/dm^3$ with the addition of sodium hypochlorite with an active chlorine content of more than 90% based on sulfur compounds. After calculating the required amount of sodium hypochlorite, we proceeded with using 10 g of sodium hypochlorite per 100 g of feedstock sample.

When using manganese dioxide as an oxidizing agent, the latter was introduced into the composite reagent in such an amount that its concentration in the leaching solution was 5.0, 10.0, and 15.0 $g/dm^3$. The experiments were carried out at room temperature.

### 3.2. Methods of Analysis

The quantitative content of nonferrous metals and iron was determined with the help of an Optima 8300 DV Optical Emission Spectrometer with inductively coupled plasma (ICP) (PerkinElmer, Norwalk, CT, USA), and for gold and silver, an AA240 Atomic Absorption Spectrometer made by Varian Optical Spectroscopy Instruments (Mulgrave, Victoria, Australia) was used.

X-ray phase data of the noble metal concentrate were obtained using a D8 Advance diffractometer (Bruker AXS GmbH, Karlsruhe, Germany) with cobalt anode and Cu K$\alpha$ radiation, and the X-ray fluorescence data were obtained on a Venus 200 (Vancouver, Canada) wave-dispersion spectrometer made by PANalytical. The diffractograms were decoded, and the interplanar distances were calculated using EVA software (HVAC version, LBS, Bhandup West, Mumbai, India). The decoding of the samples and phase search was performed using the search/match program (search/match enables you to define criteria to check for duplicate or multiple ID records) using the ASTM (American Society for Testing and Materials) Card Database.

Mineralogical analysis was performed using the A1AxioScope Microscope (Carl Zeiss Microscopy GmbH, Oberkochen, Germany). A polished section was created for mineralogical analysis from the sample material, i.e., the residue produced from the extraction of nonferrous metals and the partial extraction of iron from the activated cake. The initial materials used for the research, in addition to the residue obtained via the chemical activation of pyrite cinders and the extraction of nonferrous metals and partial extraction of iron, were pyrite cinders obtained in the process of obtaining sulfuric acid at the Tselinny Mining and Chemical Combine (now SGCC—Stepnogorsk Mining and Chemical Combine) facilities.

### 4. Conclusions

A complex processing method for pyrite cinders—man-made waste generated in the process of sulfuric acid production—was developed. This method is based on chemical activation, which contributes to the structural and phase changes in individual minerals of raw materials. The method comprises the thermal treatment of technogenic raw materials with a sodium hydrogen carbonate solution.

The optimal conditions for the process of chemical activation were determined during this study, the use of which allowed us to isolate the following in the solution: wt. %: CuO

76.8; ZnO 75.9; and $Fe_2O_3$ 26.0. The extraction degree of nonferrous metals was 15–20% lower when pyrite cinders were leached without chemical activation.

During the physical and chemical examinations of the residue containing noble metals, it was found that hematite and goethite–hydrogoethite are dominant among the ore minerals. Gold in the residue is in the ultra-fine-dispersed and fine-dispersed states, are both free and in aggregates with hematite and with goethite–hydrogoethite. Gold particles in a free state and covered with oxidation films of a goethite–limonite composition—"rusty gold"—were also found.

The process of noble metal leaching from this residue using a complex reagent based on sulfur compounds was studied. It was shown, based on the analysis of the literature data, that the choice of optimal concentrations of the components included in the complex reagent, along with the process temperature and the ore grinding degree, play a decisive role in the leaching process of noble metals.

The optimal leaching conditions for the residue obtained from the extraction of nonferrous metals and the partial extraction of iron from an activated cake by a complex reagent based on sulfur compounds and by a complex reagent in the presence of an oxidizer—manganese dioxide—were established. The composition of the complex reagent was as follows (in $g/dm^3$): $Na_2S_2O_3$, 50; $Na_2SO_3$, 100; $CuSO_4$, 2.5; S:L ratio = 1:10; and a duration of 7 h. The gold extraction degree was 87.2%, and the silver extraction degree was 75.1%. In the presence of an oxidizer, the gold extraction degree was over 90%, and that for silver was approximately 60%.

A precipitation variant of the noble metals from productive solutions with zinc dust was considered.

**Author Contributions:** Conceptualization, B.K. and T.S.; methodology, T.S.; software D.Y. and Y.A.; validation, B.K. and T.S.; formal analysis, B.K., T.S. and A.M.; investigation, T.S.; resources, Y.A.; data curation, D.F.; writing—original draft preparation, T.S. and D.Y.; writing—review and editing, T.S. and D.Y.; visualization, Y.A., A.M. and D.F.; supervision, B.K.; project administration, B.K.; funding acquisition, B.K. All authors have read and agreed to the published version of the manuscript.

**Funding:** This research was funded by the Science Committee of the Ministry of Education and Science of the Republic of Kazakhstan (Grant No. AP09259455).

**Data Availability Statement:** Data are contained within the article https://doi.org/10.31643/2021/6445.129, accessed on 22 October 2022.

**Conflicts of Interest:** The authors declare no conflict of interest.

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
