# Peer review of "On the Question of the Complex Processing of Pyrite Cinders"

_inorganics, doi:10.3390/inorganics11040171_

Round 1

Reviewer 1 Report

Dear Authors,

I would suggest the following aspects to be considered:

Title: "On the Question of the Complex Processing of Pyrite Cinders". The title is not representative of the work done. I would suggest for instance: “Pyrite Cinders Alternative Thermal Processing”.

 Keywords: “noble metals; chemical activation; thiosulfate leaching” are not suitable keywords. I suggest to remove them. I sugest to find out other keywords after modifying the title of this manuscript. Indeed, all keywords must be taken from the title of the manuscript.

 Experimental section: Provide full details (experimental conditions). Readers have to be able to reproduce all experiments and measurements.

Legends to figures and tables: Legends for all figures and tables in the manuscript should present full information about their content so that readers can understand everything about without referring to the main body of the manuscript.

Good luck

Author Response

Dear reviewer, thank you for your comments, all changes have been made to the text of the manuscript, the manuscript is attached to this letter.

Reviewer 2 Report

The paper describes the processing method for pyrite cinders with the sodium hydrogen carbonate solution. Such action causes a chemical activation of raw  materials by the structural and phase changes in individual minerals. The method facilitates the recovery of the most valuable components of pyrite cinders, which are gold and silver. Another advantage is separation of non-ferrous metals and partially iron at the first stage. The design of the experiment and structure of the manuscript is correct.

There are some corrections that should be made:

Abstract

Citations are not used in the abstract

Results and Discussion

Line 72 – L:S - what is ”L” and ”S”

Line 106 - …and chalcopyrite,  - this is the end of the phrase ?

Line 172 - Sodium hypochlorite and manganese dioxide were used as oxidants - manganese dioxide…the amount ???

Line 175 - …with concentration of 10 g/dm3 – This is not clear for me…it looks like a density…please check it

Line 177 - At the second stage the pH of pulp was adjusted to 9,5 % - remove %

Line 184 - …the concentration of the latter was 5.0, 10.0 and 15.0 g/dm3…not % ???

Line 195 - considered as g/m3 - not g/dm3 ???

Line 196 - The gold extraction degree was 87.2, and silver - 75.1% within 7 hours – missing extraction degree in Table 4 ???

Materials and Methods

Line 226 – (VP-30, «Vibrotechnik», Russia, St. Petersburg) - should be (VP-30, Vibrotechnik, St. Petersburg, Russia)

Line 227 - 1500 vpm – not rpm ???

Line 235 - Atomic Emission Spectrometer – should be Optical Emission Spectrometer

Line 239 - (Bruker AXS GmbH) – provide city and country

Line 240 - Venus 200 - provide city and country

Line 241 - …using EVA software - provide manufacturer, city and country

Line 242 - …was performed with the help of "Search/match" program – maybe more details ???

Line 244 - was performed using the A1AxioScope Microscope - provide manufacturer, city and country

Table 1 – mass fraction – units ???

Table 2, 3 -  S:L - what is ”L” and ”S” ? units should be only in the heading; dots instead of commas; concentration is written in lower case; in the table should be references…if the data comes from literature

Table 4 - dots instead of commas; concentration is written in lower case; PH…pH; temperature of the fist stage ?

Table 5 – S:L - what is ”L” and ”S” ? what is E ?

Table 6 – result – I think this column can be deleted

Figure 1 – II neutralization pH 9.7 - (should be a dot, not comma); I’m wondering if we can talk about neutralization in this case …since the final pH already exceeded 7.0…; what exactly mean solution” ???

Figure 4 – russian language on the figure …should be english language                                                                                                                       

References

Almost all the references require correction according to the journal’s requirements. References must be numbered in order of appearance in the text !

Author Response

Comments and Suggestions for Authors

The paper describes the processing method for pyrite cinders with the sodium hydrogen carbonate solution. Such action causes a chemical activation of raw  materials by the structural and phase changes in individual minerals. The method facilitates the recovery of the most valuable components of pyrite cinders, which are gold and silver. Another advantage is separation of non-ferrous metals and partially iron at the first stage. The design of the experiment and structure of the manuscript is correct.

There are some corrections that should be made:

Abstract

Citations are not used in the abstract

Results and Discussion

Line 72 – L:S - what is ”L” and ”S”- "L-liquid" and "S-solid"

Line 106 - …and chalcopyrite,  - this is the end of the phrase ? Corrected

Line 172 - Sodium hypochlorite and manganese dioxide were used as oxidants - manganese dioxide…the amount ??? It is specified in detail in the text of the article in the section "Research Methods"

Line 175 - …with concentration of 10 g/dm3 – This is not clear for me…it looks like a density…please check it with a concentration of 10 g / dm 3 - I don’t understand ... it looks like density ... please check The concentration of a substance in a solution is measured in moles (M), gram equivalents (N), and also in g / l. In the SI system, g/dm3 is used instead of g/l. (The international system of units,  SI (French Système international d'unités, SI) is a system of units of physical quantities, a modern version of the metric system. SI is the most widely used system of units in the world in science and technology.) https: // en.wikipedia.org/wiki/International_system_of_units

Line 177 - At the second stage the pH of pulp was adjusted to 9,5 % - remove % Corrected

Line 184 - …the concentration of the latter was 5.0, 10.0 and 15.0 g/dm3…not % ??? Corrected

Line 195 - considered as g/m3 - not g/dm3 ??? Corrected

Line 196 - The gold extraction degree was 87.2, and silver - 75.1% within 7 hours – missing extraction degree in Table 4 ??? The results showed that no leaching of precious metals was observed at the first stage.

Materials and Methods

Line 226 – (VP-30, «Vibrotechnik», Russia, St. Petersburg) - should be (VP-30, Vibrotechnik, St. Petersburg, Russia) Corrected

Line 227 - 1500 vpm – not rpm ??? Corrected

Line 235 - Atomic Emission Spectrometer – should be Optical Emission Spectrometer Corrected

Line 239 - (Bruker AXS GmbH) – provide city and country Corrected

Line 240 - Venus 200 - provide city and country Corrected

Line 241 - …using EVA software - provide manufacturer, city and country Corrected

Line 242 - …was performed with the help of "Search/match" program – maybe more details ??? Corrected

Line 244 - was performed using the A1AxioScope Microscope - provide manufacturer, city and country Corrected

Table 1 – mass fraction – units ??? Table 1 - mass fraction - units ??? Table 1 (new version - Table 3) shows the ratio of minerals in the residue without waste rock in%.

Table 2, 3 -  S:L - what is ”L” and ”S” ? "L-liquid" and "S-solid", units should be only in the heading; dots instead of commas; concentration is written in lower case; in the table should be references…if the data comes from literature Corrected

Table 4 - dots instead of commas; concentration is written in lower case; PH…pH; temperature of the fist stage ? Corrected

Table 5 – S:L - what is ”L” and ”S” ? what is ”E” ? "L-liquid" and "S-solid, E - extraction"

Table 6 – result – I think this column can be deleted Corrected

Figure 1 – II neutralization pH 9.7 - (should be a dot, not comma); I’m wondering if we can talk about neutralization in this case …since the final pH already exceeded 7.0…; what exactly mean ”solution” ??? « solution - this is the liquor after pulp filtration » Corrected

Figure 4 – russian language on the figure …should be english language Corrected                                                                                                                       

References

Almost all the references require correction according to the journal’s requirements. References must be numbered in order of appearance in the text ! Corrected

Reviewer 3 Report

Technologies for processing waste from the mining and metallurgical industry are in demand today. The development of technologies for a more complex extraction of valuable components into products is relevant. Therefore, the article will be of interest to readers. But the article needs to be improved.

Author Response

REVIEW 2.

Technologies for processing waste from the mining and metallurgical industry are in demand today. The development of technologies for a more complex extraction of valuable components into products is relevant. Therefore, the article will be of interest to readers. Methodically, the study was carried out correctly. Interesting results have been obtained on the effect of activation by sodium bicarbonate and oxidizing agents on the extraction of precious metals from pyrite cinders. A high recovery level has been achieved. But the article needs to be improved.

  1. The introduction does not allow us to understand why the technological approaches implemented in the study were chosen: activation, magnetic separation, leaching with manganese dioxide. The introduction is very general. Information on the use of activation with a solution of sodium bicarbonate refers to the 1936 article and does not explain why this particular reagent was chosen. The goals and objectives of the study are not specified. There is no generalization on the material of the introduction, there is no logical transition to the following sections. The introduction should be revised.

The article of 1936 includes information on the use of the method of chlorination of pyrite cinders, and not activation with a solution of sodium bicarbonate. The authors deliberately present data on the use of chlorination of pyrite cinders since 1936 to emphasize that researchers have been dealing with this problem for many years.

The development of technology for the chemical activation of pyrite cinders is based on the method

described in the work (Patent No. 32333 of the Republic of Kazakhstan, issued on July 31, 2017), This method was developed and patented by employees of the Institute of Metallurgy and Enrichment. The method contributes to the transformation of the phases of the feedstock by hydrothermal soaking it in a solution of sodium bicarbonate. According to the phase analysis, the composition of pyrite cinders includes gangue minerals: albite, dolomite and a phosphorus-containing phase, which cover and cement non-ferrous minerals. As a result of thermochemical activation, when interacting with a sodium bicarbonate solution, the waste rock phases dissolve - the dolomite phase disappears and the albite content decreases. As a result, thermochemical activation made it possible to effectively carry out acid leaching. More detailed information is presented in the introduction and text of the article.

  1. The section "Materials and Methods" is presented after the results and their discussion. It is very uncomfortable. In the section, the authors presented mainly the instrumentation of the experiment. The text does not contain a description of the object of study - pyrite cinders. There are no structural features, mineral and phase compositions. Although in another article by Kenzhaliyev B. et al. Extraction of Noble Metals from Pyrite Cinders //ChemEngineering. - 2023. - Vol. 7. - No. 1. – P. 14. Pyrite cinders were studied in detail.

The authors did not focus on the description of the object of research - pyrite cinders, which is described in detail in the article by Abikak, E.B.; Kenzhaliyev, B.K. Development of an integrated technology intended to process pyrite cinders using Chemical pre-activation // NEWS of the National Academy of Sciences of the Republic of Kazakhstan series of geology and technical sciences. 2022, Vol. 3, 32–51 pp. https://doi.org/10.32014/2022.2518-170X.178, as well as on the phase and chemical composition of the residue, in which noble metals are concentrated (Kenzhalieva B.K. et al. Extraction of noble metals from pyrite slags//Chemical Engineering. - 2023. - Issue 7. - No. 1. - P. 14), in order to avoid autoplagiarism. If necessary, you can familiarize yourself with the characteristics of these materials in these articles. Instead, the authors presented the ratio of minerals in the residue, in which precious metals are concentrated, without taking into account gangue (Table 3).

Magnetic separation parameters are not specified. The leaching technique can be understood from Tables 2-5, but this will be inconvenient for readers. There should be a description of the range of parameters for leaching using different reagents and additives of oxidizing agents. The description of the methods is scattered throughout the text of the article. They must be transferred to the "Materials and Methods" section. Corrections made to the text of the manuscript

  1. The description of the methods does not allow other researchers to reproduce the experiment. For example, there is no size of cinders, although there is a discussion about the influence of this parameter on the efficiency of leaching in line 147-149. Corrections made to the text of the manuscript
  2. It is desirable to illustrate the change in the phase composition with diagrams indicating the chemical formula of the phase before and after thermochemical activation. Corrections made to the text of the manuscript
  3. Units of measurement in table. 1 are not specified. The content of gold in the leaching residue of 0.3% raises doubts. Probably 0.3 g/t?

Table 1 (in the corrected version of Table 3) shows the ratio of minerals in the residue, excluding waste rock, in%. The gold content in the residue was 1.9 g/t, silver - 16.9 g/t.

  1. There are no compositions of productive solutions. Including the composition of the solution from which the cementation of gold on zinc dust is carried out. Corrections made to the text of the manuscript
  2. Figure 3 has very poor quality photographs. The photographs in Figure 3 have been replaced by the data in Table 4.
  3. If the text contains an indication that “Based on the analysis of the scientific literature, the following compositions of the complex reagent were selected” (line 154), then it is necessary to make references to this literature. Corrections made to the text of the manuscript

The literature cited by the authors is mostly old. The bibliography needs to be updated. The list of used literature has been updated and expanded.

Round 2

Reviewer 3 Report

The authors reasonably explained their position on some issues. The article has undergone significant changes in accordance with the comments of the reviewer. The bibliography has been updated. Methods and research results are clear. The photo quality has been improved. I was interested in reading the article. Good luck to the authors in their future research.

Author Response

Dear Reviewer,

thank you very much for your contribution to improving our research article!

Kind regards,

Authors